# Assessment of Ferritic ODS Steels Obtained by Laser Additive Manufacturing

**DOI:** 10.3390/ma16062397

**Published:** 2023-03-16

**Authors:** Lucas Autones, Pascal Aubry, Joel Ribis, Hadrien Leguy, Alexandre Legris, Yann de Carlan

**Affiliations:** 1Service de Recherches Métallurgiques Appliquées, Université Paris-Saclay, CEA, 91191 Gif-sur-Yvette, France; 2Service d’Études Analytiques et de Réactivité des Surfaces, Université Paris-Saclay, CEA, 91191 Gif-sur-Yvette, France; 3Unité Matériaux et Transformations, Université de Lille, 59655 Villeneuve d’Ascq, France

**Keywords:** additive manufacturing, Powder Bed Fusion (PBF), ferritic steels, Oxide Dispersion Strengthened (ODS) steels, Small Angle X-ray Scattering (SAXS), transmission electron microscopy (TEM)

## Abstract

This study aims to assess the potential of Laser Additive Manufacturing (LAM) for the elaboration of Ferritic/Martensitic ODS steels. These materials are usually manufactured by mechanical alloying of powders followed by hot consolidation in a solid state. Two Fe-14Cr-1W ODS powders are considered for this study. The first powder was obtained by mechanical alloying, and the second was through soft mixing of an atomized Fe-14Cr steel powder with yttria nanoparticles. They are representative of the different types of powders that can be used for LAM. The results obtained with the Laser Powder Bed Fusion (LPBF) process are compared to a non-ODS powder and to a conventional ODS material obtained by Hot Isostatic Pressing (HIP). The microstructural and mechanical characterizations show that it is possible to obtain nano-oxides in the material, but their density remains low compared to HIP ODS steels, regardless of the initial powders considered. The ODS obtained by LAM have mechanical properties which remain modest compared to conventional ODS. The current study demonstrated that it is very difficult to obtain F/M ODS grades with the expected characteristics by using LAM processes. Indeed, even if significant progress has been made, the powder melting stage strongly limits, for the moment, the possibility of obtaining fine and dense precipitation of nano-oxides in these steels.

## 1. Introduction

Ferritic or martensitic (F/M) Oxide Dispersion Strengthened (ODS) steels typically contain an ultra-fine (1 to 5 nm in radius) and homogeneous dispersion of thermally stable nano-oxides Y-Ti-O [1,2]. They have been originally developed for nuclear applications to improve the high-temperature properties of body-centered stainless steels [3,4]. The body-centered cubic lattice provides excellent resistance to irradiation-induced swelling compared to the face-centered cubic lattice, while the high density of nano-oxides inside the matrix significantly improves their high-temperature mechanical properties, especially the creep strength [5,6]. Small nano-oxides can also act as sink sites for He formed by detrimental transmutation in the material under irradiation [7]. These properties make ODS steels excellent candidates for fuel cladding tubes in Sodium Fast Reactors (SFRs) or structural materials in fusion reactors. Indeed, these components could undergo irradiation damage up to 200 dpa and operate in a temperature range of 400–800 °C [8,9].

ODS steels are prepared through powder metallurgy. The nano-oxides dispersion in the steel is usually obtained with high-energy milling, called mechanical alloying (MA), of an as-atomized steel matrix powder and a reinforcement oxide powder, such as Y_2_O_3_. During MA, Y_2_O_3_ is dissolved, leading to a near-solid solution of Y and O within the highly deformed matrix powder particles [10,11,12]. Mechanical alloying is followed by hot consolidation, Hot Isostatic Pressing (HIP), or Hot Extrusion (HE), during which densification and precipitation take place.

This conventional route has been widely studied and has shown its performance to provide good quality ODS steels. However, this entire process involves numerous complex steps. In addition, the final part of geometries that can be obtained is limited, while the first wall of fusion reactor blankets is expected to contain internal cooling channels. In this context, several teams around the world are studying alternative routes that could offer new possibilities, often trying to avoid the costly MA step [13] or by studying new means of consolidation [14,15,16].

Recent technological developments in laser additive manufacturing (LAM) techniques now enable the production of parts with complex geometries, layer by layer. These processes have many advantages over conventional manufacturing methods and can be combined with them to improve manufacturing [17]. They also offer new freedom of design and the opportunity to tailor the microstructure along with the part geometry [18]. During the last decade, a few authors have used MA ODS powder as a raw material to produce ODS steels by Selective Laser Melting (SLM) or Laser Metal Deposition (LMD). These two processes are based on the powder laser fusion by a focused beam and its rapid solidification. In SLM, the layers of powder are successively spread on the previously solidified layer under an inert atmosphere chamber. It is a Powder Bed Fusion process (PBF). In LMD, an inert carrier gas directly projects the powder into the melt pool created by the laser beam. It is a Direct Energy Deposition process (DED).

Walker et al. were the first to use a PM2000 mechanically alloyed ODS powder in SLM [15], followed by Boegelein et al. [19,20]. These authors demonstrated that it was possible to retain a relatively fine distribution of nano-oxides of approximately 30 nm in size within the grains, despite the powder fusion. Boegelein et al. showed that it was possible to obtain a room temperature strength close to that of a recrystallized ODS PM2000 [19]. However, Hunt et al. and Vasquez et al. highlighted some limitations in using a mechanically alloyed ODS powder with the SLM process [21,22,23]. The usually coarse size of the powder makes it more difficult to achieve a high density of parts with classical process parameters, and tensile properties are lower than conventional ODS steels at room temperature. Other authors have used mechanically alloyed ODS powder in LMD [24,25,26,27]. They obtain microstructural characteristics similar to those obtained in SLM, with coarse grains and distribution of nano-oxides with an average size of about 50 nm inside the grains.

This study aims to assess the elaboration of ferritic ODS steels by LAM, and three types of powder were selected:A reference Fe-14Cr-1W unreinforced powder,A conventional Fe-14Cr-1W ODS powder obtained by mechanical alloying,A “nanocomposite” Fe-14Cr-1W ODS powder obtained by TURBULA^®^ mixing.

The alternative “nanocomposite” ODS powder allows the yttria to be homogeneously dispersed on the powder particle surface, avoiding the costly mechanical alloying stage, and preserving the spherical shape and optimum size of the initial powder particles. The purpose of this work is to determine if using such “nanocomposite” powder enables ferritic ODS steels with a finer and denser nano-oxide population comparable to conventional ODS steels. The materials are consolidated by SLM from each ODS powder and from the unreinforced steel powder. A material consolidated by HIP from the mechanically alloyed ODS powder is also used as a reference ODS steel. The microstructure, nanoprecipitation, and high-temperature tensile properties are characterized and compared in order to assess ODS steel manufacturing by LAM. The powders used in this study can be considered quite representative of the diversity of powders that can be used in LAM to obtain F/M ODS materials; a powder where yttrium and oxygen are in solid solution (MA-R0.2) and a powder where yttrium and oxygen are introduced in the form of a gangue around initial powders (SM-R0.5).

## 2. Materials and Methods

### 2.1. Powder Analysis

A pre-alloyed Fe-14Cr-1W-0.22Ti stainless steel atomized powder, labeled A-Ti, was supplied by Nanoval (Berlin, Germany) and used as a matrix material to produce two ODS powders. The ODS powder obtained by mechanical alloying is labeled MA-R0.2 and was milled by the company Plansee (Reutte, Austria) with 0.2 %wt of Y_2_O_3_ powder. The powder MA-R0.2 was sieved at 100 μm after milling. The detail of the milling parameters lay under industrial confidentiality but is known to produce good ODS [8]. The nanocomposite ODS powder obtained by soft mixing is labeled SM-R0.5 and was mixed with 0.5 %wt of Y_2_O_3_ nanoparticle powder (<50 nm) in a three-dimensional mixer (TURBULA^®^, WAB group) for 7 h. The A-Ti powder was sieved at 50 µm before mixing, and AISI steel balls with a diameter of 5 mm were added to the container (1 L) with a balls-to-powder ratio of 1:1. These steel balls enabled the deagglomeration of the Y_2_O_3_ nanoparticles during mixing. The filling rate of the container is approximately 40%. Finally, an unreinforced Fe-14Cr-1W stainless steel atomized powder, labeled A, was supplied by Nanoval and is used as a reference material. The chemical compositions of the powders reported in Table 1 were measured by plasma emission spectroscopy (ICP) and infrared absorption.

Figure 1 shows scanning electron microscope (SEM) images of the A-Ti, SM-R0.5 and MA-R0.2 powder particles. Table 2 shows the size distribution of each powder, measured with a Partica LA-950 laser particle size distribution analyzer from Horiba^®^. Soft mixing does not deform the powder particles and preserves the size and spherical shape of the original A-Ti powder. After mixing, Y_2_O_3_ nanoparticles are homogeneously attached to the surface of the matrix powder particles.

### 2.2. Laser Additive Manufacturing

The consolidation of the powders by SLM is carried out on the SAMANTHA platform at CEA Saclay, with a TruPrint series 1000 machine (TRUMPF GmbH) equipped with a Yb laser fiber (λ = 1.064 μm) 200 W and a beam size of 55 μm. A constant flow of argon gas ensures the inert atmosphere of the chamber achieves oxygen concentrations lower than 1000 ppm during consolidation. The consolidated samples are 8 mm × 12 mm × 14.5 mm in dimension on a 316L substrate, as shown in Figure 2. The laser scans each layer of these samples following parallel lines separated by a hatch distance (HD). After each layer, the lines are rotated by 90°. Table 3 reports the parameter ranges used to consolidate the different powders. Parameters have been chosen following three objectives: relatively high ratio *P_Laser_*/*V_Laser_*, maintaining an acceptable construction time, and obtaining an acceptable density for basic microstructural and mechanical analysis. For atomized and ODS nanocomposite powders (A, A-Ti, SM-R0.5), the optimized parameters used to consolidate these powders are a laser power of 175 W, a laser scanning speed of 800 mm.s^−1^, a hatch distance of 80 µm, and a layer thickness of 30 µm. For the mechanically alloyed ODS powder (MA-R0.2), the optimized parameters used to consolidate this milled powder are a laser power of 175 W, a laser scanning speed of 300 mm.s^−1^, a hatch distance of 90 µm, and a layer thickness of 20 µm. Unless stated otherwise, microstructural and mechanical characterizations are carried out on samples consolidated with optimized parameters.

The laser volume energy density (*VED*) corresponds to the amount of energy delivered to the material per unit volume and is defined here by Equation (1):(1)$$VED=\frac{P_{Laser}}{V_{Laser}.d_{Laser}.\Delta Z}\;\left(J.mm^{-3}\right)$$
where *P_Laser_* is the laser power (W), *V_Laser_* is the laser scanning speed (mm.s^−1^), *d_Laser_* is the diameter of the laser beam (µm), Δ*Z* the layer thickness of the powder spread on the building platform (µm). The *VED* can be useful for comparing properties like the density of parts consolidated with different parameters. Another definition of Equation (1) considering the hatch distance (HD) instead of the beam diameter *d_Laser_* is often used in SLM-related literature. With powders A, A-Ti, and SM-R0.5, HD varies from 65 µm to 120 µm to consider the wide variation of the *P_Laser_*/*V_Laser_* ratio. Within the parameter range applied, the definition given in Equation (1) was preferred in this study.

However, this notion should be handled with care. Whatever the definition, the *VED* does not enable a description of the physics and geometry of the melt pool perfectly, and it does not consider the possible effects of different laser scanning patterns.

### 2.3. Experimental

The density of SLM samples is measured with the Archimedes method. The relative densities displayed in this study are calculated as the ratio between the SLM and HIP (MA-R0.2) materials (as the HIP material is well known to have a very low porosity).

Optical microscopy images are taken on a Reichert-Jung MeF3A microscope from Leica. The samples were mechanically polished, then etched with Villela reagent to reveal grain boundaries and molten pools.

Scanning electron microscope (SEM) images of the powders are taken on a VEGA3 TESCAN microscope equipped with a LaB6 gun with a voltage of 30 kV.

The samples are analyzed in a plane parallel to the building direction. They are mechanically polished following a standard metallography procedure for stainless steel, with a final chemical polishing step using a solution of silica particles in colloidal suspension to reveal the microstructure. The observations are carried out on a Zeiss Sigma HD Scanning Electron Microscope (SEM) at a voltage of 15 kV. For particle size analysis, high-definition images (4096 × 3072 pixels) are acquired with a backscattered electron detector (BSE), which allows resolutions of approximately 5 nm per pixel. Particle size (equivalent sphere radius) analysis is extracted using a machine learning plugin, Trainable Weka Segmentation, integrated into the ImageJ software [28]. The software determines the surface of the particles, and they are assumed to be spherical to deduce their radius. This method was used to characterize swelling bubbles in irradiated AIM1 austenitic steel [29].

Electron backscattering diffraction (EBSD) analyses are performed to determine the morphology of the grains and their orientation in the samples. The same Zeiss Sigma HD SEM is used with a voltage of 20 kV. Data are acquired on Esprit Bruker software from a Bruker e-Flash HR detector and are postprocessed with OIM Analysis software from EDAX. The step of the EBSD maps is 0.47 µm. Postprocessing is performed with a grain angle tolerance (GTA) of 5°. Mechanically polished specimens are electropolished before EBSD using a perchloric acid solution to remove irregularities and the deformation layer on the surface.

Thin specimens are prepared for the transmission electron microscope (TEM) by mechanical polishing using SiC grinding discs in order to refine material slices to approximately 100 µm. Discs of 3 mm in diameter are punched into the refined slice, which are then finally thinned for electronic transparency by electropolishing using a Tenupol 5 twin-jet electropolisher from Struers. The electropolishing voltage is 30 V, the electrolyte is a solution of 10 vol% of perchloric acid in ethanol, and the temperature is maintained at 10 °C. TEM observations were conducted on a Cs-corrected (probe and image) JEOL-neoARM microscope operating at 200 keV and equipped with annular dark field and bright field detectors for STEM acquisition. The microscope is also equipped with double Centurio EDS detectors from JEOL.

Small Angle X-ray Scattering (SAXS) experiments have proven to be a powerful method to identify and characterize dense populations of nano-sized particles in a metal matrix, particularly in ODS steels [30,31,32]. In this study, SAXS measurements were performed on a laboratory setup at CEA. The energy of the X-ray source (Mo rotating anode) is 17.44 keV, with a beam diameter of ~0.8 mm. The sample thickness is mechanically polished down to 60 to 80 µm to achieve a suitable transmission regarding the energy used. Scattering patterns were acquired on a 2D detector, which enables it to cover a wide range of scattering vector *q*. The detector-to-sample distance was set to 70 cm, with an acquisition time of approximately 4 h. Each 2D scattering pattern was azimuthally integrated, background-corrected, normalized by incident beam intensity, sample transmission, thickness, and solid angle viewed by the detector. The intensity was reduced to absolute units thanks to a glassy carbon standard.

The SAXS data were then fitted with SASview software [33]. The fitting procedure uses a simulated pattern from a lognormal distribution of spherical precipitates with a dispersion fixed at a value of 0.1. It also includes a power-law contribution *A* × *q^-n^*, known as the Porod contribution of large microstructural features (coarse precipitates, porosities). A constant contribution is also considered, arising from other factors, such as the matrix solid solution. After fitting, the output data are then the power law parameters (A and n), the Laue constant, the mean radius, and the volume fraction of the scattering particles. The volume fraction cannot be extracted without knowing the electronic contrast Δ*ρ* (difference in scattering length density) between the particles and the matrix. The scattering length density depends on the incident beam wavelength, the chemical composition, and the atomic volume. In the present study, precipitates are assumed to be Y_2_Ti_2_O_7_ pyrochlores, which a reasonable hypothesis on fully consolidated ODS steel, and is consistent with previous SAXS studies [30,32]. The scattering length density values for the Fe-14Cr matrix and the Y_2_Ti_2_O_7_ particles are 62.4 × 10^−6^ Å^−2^ and 38 × 10^−6^ Å^−2^, respectively. They are determined thanks to the SLD calculator available in SASview. More details on the SAXS equations and fitting methods applied to ODS steels can be found in SASview documentation or the literature [34,35].

The mechanical tensile tests are carried out on specimens with a gauge length of 6 mm and a section of 1 mm × 1.5 mm. All the specimens are machined in SLM blocks in the direction parallel to the building direction (BD). The tests were carried out in the air at temperatures ranging from 20 °C to 700 °C. The strain is controlled with a strain rate of 7.10^−4^ s^−1^.

## 3. Results

### 3.1. Effect of the Powder and the Laser Parameters on the Relative Density

The density of SLM parts is compared to that of an ODS steel obtained by Hot Isostatic Pressing (HIP) from the mechanically alloyed powder MA-R0.2. The HIP part contains very little porosity and has a density of 7.723 g.cm^−3^. A first optimization of the parameters is carried out on the raw atomized powder A-Ti and on the milled powder in order to achieve sufficient densities and estimate the melt pool dimensions. Figure 3 shows the relative densities of the parts consolidated by SLM from the different powders as a function of the *VED*.

The consolidations of the matrix powder (A-Ti) and nanocomposite ODS powder (SM-R0.5) are realized with the same set of parameters. Densities greater than 98% are achieved with a *VED* lower than 150 J.mm^−3^. Consolidation of the coarser milled powder requires different sets of parameters. Densities close to 98% can be achieved with higher *VED* (>400 J.mm^−3^) and $\frac{P_{Laser}}{V_{Laser}}$ ratio (>0.55), as shown in Figure 3. This is because larger particles necessitate higher energy to melt completely.

Adding Y_2_O_3_ satellite particles to the surface of the atomized matrix powder can decrease the flowability of the powder during layering and change the absorbance of the laser radiation. This can therefore have a detrimental effect on the final density of parts. Nevertheless, in this study, no significant differences are observed between the density of parts consolidated with the raw atomized powder and with the nanocomposite ODS powder. At 175 W, the density of parts consolidated with nanocomposite ODS powder decreases very slightly compared to parts consolidated with raw atomized powder. With the optimized parameters (91 J.mm^−3^), the density reduces from 98.8% to 98.5%. Thus, the addition of 0.5 %wt of Y_2_O_3_ seems to have little impact on the absorbance of the laser by the powder.

Using a spherical and finer powder enables dense parts with a wider range of parameters. Therefore, soft mixing makes it possible to manufacture parts by SLM with increased flexibility compared to powders from mechanical alloying. In the case of ODS steels, the choice of laser parameters must also be made considering the microstructure and the precipitation of Y-Ti-O nanoparticles. Parameters leading to high cooling rates will enable better retention of the yttrium in the matrix and reduce the growth of oxides formed in the bath. Due to the shape of the powder, a compromise between good density and a high cooling rate is, therefore, much easier to achieve with the ODS powder SM-R0.5 rather than the ODS powder MA-R0.2.

### 3.2. Microstructural Characterizations

The microstructures of the SLM materials consolidated with optimized parameters for each powder are analyzed in the plane parallel to the building direction (BD). Figure 4 presents optical images after etching with Villela, pointing out the melt pools on the last layer of the parts and their respective grain structure. The dimensions of the melt pools were measured from this type of optical image after cutting transversely to the laser lines of the last solidified layer. The average values are presented in Table 4 and were obtained after measuring 15 to 20 melt pools. It appears that the melt pools of ODS steels are shallower and wider than those of unreinforced Fe-14Cr-1W steel.

All materials have a columnar microstructure, as expected in SLM. The material consolidated from the unreinforced powder A presents differences compared to ODS parts, with columns well identified by the laser lines on which grains are superimposed (Figure 4a). They are less elongated, with a length of around 100 to 500 µm, and have a width between 70 and 100 µm. ODS parts consolidated from SM-0.5 and MA-0.2 powders exhibit very similar microstructures, dominated by long columnar grains from 20 to 100 µm in width and several hundred microns in length, some even up to a millimeter (Figure 4b,c).

EBSD analyses are carried out along the building direction. The texture of the material consolidated from the unreinforced (A) is less clear compared to ODS parts, mostly alternating between grains <100>//BD and <111>//BD (Figure 5a). In addition, EBSD mapping shows numerous finer grains between the columnar grains. ODS steels exhibit a strong <100>//BD crystallographic texture, as shown in Figure 5b,c, which corresponds to the easy growth direction for the bcc metal matrix [36]. ODS steel consolidated with the nanocomposite powder exhibits greater disorientations between the columnar grains.

### 3.3. Analysis of the Precipitation

#### 3.3.1. Coarse Phases

Figure 6a,b shows coarse phases present in both ODS steel consolidated with powders SM-R0.5 and MA-R0.2, respectively. These strip-like coarse phases are around 5 to 30 µm in size, and EDX analyses on these phases summarized in Table 5, show that they are mostly constituted of yttrium. These yttrium-rich phases are often located above the last solidified layer on the top of the samples, as shown in Figure 6a,b, although several are observed trapped inside the part. This suggests that these phases are formed by the agglomeration of yttrium during melting and then float out of the melt pool, as is the case with casting processes [37]. These phases are slag-like and deplete the yttrium available for nanoprecipitation, which is likely to be detrimental to the mechanical properties of the alloy, such as strength and fatigue resistance.

#### 3.3.2. Analysis of the Nanoprecipitation with SEM

The dispersion of the nanoprecipitates is first assessed by high-resolution SEM, which enables the observation of particles with a size between 10 to 100 nm or larger over a wide area compared to TEM.

Figure 7a shows the distribution of nanoparticles in the material produced from the unreinforced powder A. Figure 7b,c shows the distribution of nanoparticles in the ODS materials produced by SLM, respectively, with powders SM-R0.5 and MA-R0.2. The unreinforced ferritic steel contains a dispersion of nanoprecipitates similar to those in ODS parts. In each material, the nanoprecipitates are homogeneously distributed within the grains. EDX spectra on the coarser nanoparticles in ODS parts show that they contain at least titanium and sometimes yttrium.

The nanoprecipitate size distribution in both ODS materials is obtained from several SEM images acquired in different areas of each part, which allowed a count of over 2500 particles. They are compared in Figure 8. The nanoparticle mean radius and area density in the ODS steel consolidated from powder MA-R0.2 are 27.5 nm and 3.43 × 10^12^ m^−2^, respectively. The ODS steel consolidated from powder SM-R0.5 has a finer and denser population of nanoparticles with a mean radius and area density of 14 nm and 1.00 × 10^13^ m^−2^, respectively. Moreover, almost no particles larger than 30 nm in radius are observed with the nanocomposite ODS powder. TEM and SAXS are required to analyze smaller precipitates.

#### 3.3.3. Analysis of the Nanoprecipitation with TEM

The dispersion of the finest nano-oxides is observed in TEM on a thin specimen extracted from the SLM material consolidated with powder SM-R0.5, as SEM analysis suggested that this material has the finest nano-oxide distribution. Several authors have already studied SLM materials consolidated from milled ODS powders in TEM [15,20,22,25].

Figure 9 shows the distribution of nanoparticles observed in TEM. The nanoparticles are mostly distributed homogeneously in the grains and not at grain boundaries, which is consistent with the SEM observations. Numerous dislocations seem to be pinned by the nanoprecipitates. As the histogram in Figure 10 shows, two size ranges of nano-oxides appear; one centered around 5 nm in radius and one centered around 15 nm in radius, observable by SEM. The total number of particles counted in TEM was 371, for a nanoparticle area density of 1.67 × 10^14^ m^−2^ and a mean radius of 8.33 nm. The depth of analysis was approximately 100 nm according to EELS measurement. Particles with a radius smaller than 5 nm were difficult to analyze due to contrast problems caused by a large number of dislocations and due to their low density in the material. Nevertheless, Figure 11 shows a precipitate of about 2 nm in diameter. Although the nature of such a small particle is difficult to determine, the FFT seems to indicate that it has a cubic structure and an orientation relationship with the matrix.

The chemical nature of the nanoparticles was studied using EDX analysis in TEM. The EDX map displayed in Figure 12 shows a large number of titanium-rich nano-particles and only one particle containing titanium and yttrium. The observed nanoparticles are therefore constituted of different natures of nano-oxides (at least titanium oxides and Y-Ti-O oxides). It is difficult to assess the predominant nature of nano-precipitates with TEM observations, although titanium oxides appear to be in the majority of the ODS material consolidated with the powder SM-R0.5. This is consistent with the analysis performed with EDX spectra in SEM. In order to have further insights regarding the nature of these nanoprecipitates, chemical analyses were realized on the consolidated SLM parts. Powder SM-R0.5 contains 0.3 %wt of yttrium before SLM (Table 1), while the consolidated material after SLM only contains 0.05 %wt of yttrium, as shown in Table 6. Despite the Y-rich slag phases and the Y-Ti-O nano-oxides detected, yttrium seems to be in low amount in this ODS material after SLM. This loss of yttrium does not occur in the ODS steel consolidated with MA-R0.2 powder.

### 3.4. Small Angle X-ray Scattering

SAXS measurements of SLM parts are compared to that of a conventional ODS steel obtained by HIP from the ODS powder MA-R0.2, as shown in Figure 13. Such a characterization method is well known to study the nanoparticles of conventional ODS but has never been used on ODS steels obtained by additive manufacturing. The arrow on this HIPed ODS SAXS curve points out the region where the scattering from the distribution of nanoparticles (i.e., the nanoparticle contribution) is visible. The fitted mean radius and volume fraction in this material are 2.0 nm and 0.23%, respectively, resulting in a nano-oxide density of 6.7 × 10^22^ m^−3^. The nanoparticle contribution is not visible in any of the SLM parts curves, which display only a Porod and constant contributions (mostly arising from large features and solid solutions, respectively). This indicates that the volume fractions and/or electronic contrast of nanoparticles are low in those materials. Assuming that the nanoparticles are Y_2_Ti_2_O_7_ pyrochlores (demonstrated as wrong by EDS but given as a comparison point), this leads to a volume fraction much lower than 0.1% for both unreinforced and ODS steels obtained after SLM.

### 3.5. Tensile Tests

The tensile tests are performed at room temperature, 650 °C, and 700 °C. Yield strengths are measured at 0.2% plastic strain. To properly evaluate the ODS materials obtained by SLM, their mechanical behavior was compared to the HIP ODS material and the non-ODS material obtained by SLM (Figure 14). The stress-strain curves of the HIP ODS and the SLM materials consolidated with powders A, MA-R0.2, and SM-R0.5 are shown in Figure 14a–d, respectively.

The SLM ODS elaborated from powder MA-R0.2 has a brittle behavior. At room temperature, rupture occurs in the elastic regime around 300 MPa, whereas the materials consolidated with powders A and SM-R0.5 have a ductile behavior and yield strength of 411 MPa and 397 MPa, respectively. The HIP ODS material has mechanical properties at high temperatures that are clearly superior to the non-ODS material obtained by additive manufacturing. The ODS obtained by LAM also remains much less efficient than the HIP ODS. Even if the mechanical properties of the SM-R0.5 ODS are more interesting than the MA-R0.2 ODS, they are comparable to the non-ODS material. The addition of Y_2_O_3_ particles by mechanical alloying or soft mixing does not improve the mechanical properties compared to an unreinforced steel powder after SLM.

## 4. Discussion

The powder properties, such as its size, shape, surface morphology, amount of internal porosity, and chemical composition, have a strong impact on the LAM processes [38,39]. Mechanical alloying dissolves the yttrium in the matrix powder and introduces a high density of dislocations for precipitation in the solid state. However, laser melting destroys this favorable metallurgical state for the conventional route. Since mechanical alloying is an expensive, complex step and loses its value in laser additive manufacturing, the development of alternative ODS powders for such processes appears relevant. The “nanocomposite” powder elaboration method presented in this study enables the preservation of the size and spherical shape of the initial atomized steel powder, and yttrium is introduced with Y_2_O_3_ nanoparticles distributed homogeneously on the surface of the atomized powder particles.

A few recent studies have also focused on the development of similar ODS powders. Gao et al. use such a type of Fe-18Cr-2W-0.5Ti-0.3Y_2_O_3_ powder in electron beam melting (EBM) after obtaining it by soft ball-milling [40]. Doñate-Buendía et al. obtain a similar powder of composition PM2000 thanks to a process they call laser fragmentation in liquid, which they use in SLM and LMD [41]. Several authors have also developed austenitic ODS steels in SLM from powders obtained by low-energy milling in an attritor or a ball mill [42,43,44]. Such “nanocomposite” powders are also used for different materials or with other natures of nano-reinforcements, and their development is particularly dynamic for metal matrix composite materials (MMC) [45,46]. Other authors, such as Jia et al., proposed a fabrication strategy of ODS using in situ synthesis of nanoparticles during printing with a gas atomized pre-alloyed powder containing yttrium [47].

In this study, the microstructure formation in MA-R0.2 and SM-R0.5 ODS obtained by SLM was analyzed. These materials are representative of what is observed in the literature.

### 4.1. Effect of the Powder on the Solidification Microstructure

Chemical composition determines key parameters, such as surface tensions of the melt pool in LAM processes. Table 4 shows that the ODS materials have a larger and shallower melt pool shape than the Fe-14Cr-1W steel, which is narrower and deeper and looks like a pre-keyhole mode. Vasquez et al. also observed this result [23]. Several authors have studied the influence of alloying elements on the shape of the melt pool.

The convective flows of the liquid within the melt pool are known as the Marangoni flow. They are due to the surface tension gradient between the center and the edges of the bath. Since the surface tension is temperature dependent, this gradient is itself caused by the temperature gradient which prevails in these regions because of the Gaussian distribution of the incident laser beam [48]. It is well known that alloying elements, even in small amounts, can modify the surface tension [39]. By amplifying the Marangoni flows from the center to the edges of the bath, surface-active elements (such as O, N, and sulfur) could therefore contribute to the enlargement of the melt pool. Sulfur is often present in steels and is known for inducing such an effect. However, it was not analyzed in the powders of this study. The Ni and Mo measured in the Fe-14Cr-1W steel could be responsible for a difference in surface tension compared to the ODS steel (Table 1). Deoxidizer elements (Ti or Si, for example) present in larger amounts in ODS (mostly Ti and Y) could also modify the influence of oxygen on the surface tension compared to the unreinforced steel. Other authors suggest that the presence of deoxidizer elements leads to the expansion of the melt pool due to exothermic reactions during the formation of oxides or carbides and to a tendency to ball [49].

Despite the different laser parameters, the solidification microstructure obtained with the ODS powder SM-R0.5 is very similar to that obtained with the ODS powder MA-R0.2 (Figure 5b,c). These two powders, with very similar chemical composition, led to a microstructure that is relatively different from that obtained with the unreinforced steel powder A (Figure 5a), yet consolidated with the same SLM parameters as powder SM-R0.5. These different solidification microstructures are likely due to the difference in the shape of the melt pool. Its curvature and orientation between the successive layers can have a strong influence on the orientation of the predominant grains [50,51]. In a more flattened bath, as in the ODS materials in this study, the solidification cells are mainly oriented along the building direction. In a deeper bath, such as in the Fe-14Cr-1W steel, the solidification cells on the sides of the bath will be more likely to be oriented perpendicularly to the building direction, leading to greater competition between the columnar grains formed and their orientation.

### 4.2. Yttrium Loss and Coarse Phases Issues

Yttrium has very low solubility in iron, even at high temperatures [52]. Non-equilibrium processes, such as mechanical alloying or rapid solidification, can enable the retention of such an immiscible system under a single phase in a solid solution. However, melting processes are now known to be detrimental to the manufacture of ODS steels. They result in strong agglomeration and coarsening of yttrium oxides, with an inhomogeneous distribution and the presence of slag phases [37]. Observations made in this study seem to show that a homogeneous liquid solution is not achieved during PBF. With both ODS powders, slag phases rich in yttrium are present inside and at the surface of the parts (Figure 6). Similar coarse phases have been observed in other studies with both mechanically alloyed powder or nanocomposite powder [20,44,53]. EDX analyses on these phases provided in Table 5 show a deviation from Y_2_O_3_ stoichiometry (~90 %wt of Y and ~ 4 %wt of O against 79 %wt of Y and 21 %wt of O for Y_2_O_3_). This seems to be supported by the brighter contrast of these phases in SEM-BSE observations compared to the matrix. Other alloying elements, such as titanium, are also present in an enriched content relative to the matrix (1 to 5 %wt). Y_2_O_3_ satellite particles in powder SM-R0.5 would then be melted before the yttrium agglomerates again with oxygen and other alloying elements. Based on a molecular dynamic simulation performed by Alvarez et al., Y_2_O_3_ can lose its stoichiometry from 1500 K to up to 15% in oxygen loss at high temperatures [54]. Ghayoor et al., indicate that the loss of oxygen can lead to a drop in the melting point on the Y-O diagram. The melting point of Y_2_O_3_ (2430 °C) could drop below the maximum temperature predicted in austenitic stainless steel in SLM (2200 °C) [44]. The large surface area to volume ratio of the initial Y_2_O_3_ nanoparticles can also promote their melting.

The similarities of these phases with both ODS powders SM-R0.5 and MA-R0.2 suggest that their formation mechanism is the same and support the idea that yttrium is in a dissolved state in the melt pool. These phases solidify before the matrix, and their elongated shape indicates that they could form at the surface of the melt. Although having a slightly different composition from Y_2_O_3_, they can migrate or remain at the surface of the melt due to buoyancy caused by the difference in density between Y_2_O_3_ (5.01 g/cm^3^) and Fe-14Cr-1W steel (~7.8 g/cm^3^). Marangoni flows and partial melting of previous layers could explain why some of these slag phases remain inside the part. Y_2_O_3_ has been shown to have poor wettability with steel (contact angle on Fe-Cr steel measured at 110°) [55]. These phases can thus destabilize the melt pool by increasing the surface tension locally and promote a balling effect [44]. In addition, they can prevent good bonding between two solidified layers due to their poor wettability with the liquid matrix. These slag phases can therefore deteriorate the density and the quality of the parts, in addition to the mechanical properties, as Vasquez et al. observed [23].

Table 6 shows a large loss of yttrium in the SLM part consolidated with powder SM-R0.5 (0.3% in the powder and 0.05% in part). This result has been observed in the thesis work of Kini with a similar powder obtained by soft-milling [53]. The yttrium content was measured before and after LMD and SLM for compositions containing 0.5%, 2%, and 5% of Y_2_O_3_. The amount of yttrium in the powder is comparable to the targeted value before LMD or SLM. After LMD, the yttrium content ranges from 0.05% to 0.11%, depending on the initial composition. The yttrium content remaining after SLM is approximately 0.15% for an initial content of 0.5%, which is more than in LMD or in our study, but still a considerable loss. The SLM part consolidated with the milled powder MA-R0.2 does not show such a loss of yttrium (0.146% in the powder and 0.12% in part). The reasons causing the loss of yttrium after LAM are not clear, and no mechanism has yet been provided. The observed slag phases contribute to the yttrium measured in part by ICP and deplete the amount of yttrium available for the formation of nanoparticles whatever the type of ODS powder.

### 4.3. Nano-Oxide Precipitation

In order to better retain yttrium in a solid solution and limit the growth of the first particles formed in the melt pool, high cooling and solidification rates are preferred with LAM processes to manufacture ODS. In SLM, the cooling rate is higher (10^5^ to 10^6^ °C.s^−1^) than in LMD (10^2^ to 10^4^ °C.s^−1^) [39,56]. It is also known that reducing the $\frac{P_{Laser}}{V_{Laser}}$ ratio increases the cooling rate in LAM [39]. The influence of the cooling rate on precipitation is particularly visible in Figure 8. The use of a lower $\frac{P_{Laser}}{V_{Laser}}$ ratio, made possible with powder SM-R0.5, induces finer particles. LMD process would also lead to greater coarsening of nano-oxides, as Euh et al. point out [25]. This was verified on such material obtained by LMD with powder MA-R0.2.

However, the STEM-EDX map (Figure 12) shows very few nanoparticles containing Y but a large number of nanoparticles containing Ti. This seems to indicate that yttrium contributes poorly to the nanoprecipitation in the material consolidated with powder SM-R0.5. Although the predominant nature of nanoprecipitates is difficult to assess with TEM observations alone, this result is supported by chemical analysis (Table 6), showing that very little yttrium remains after SLM and that it tends to form coarse phases (Figure 6). On the contrary, several authors report identifying in TEM only Y-Ti-O (or Y-Al-O for PM2000) oxides in SLM parts consolidated from a mechanically alloyed ODS powder [15,20,22]. The SLM part consolidated with ODS powder MA-R0.2 was not observed in TEM in this study, but these results remain consistent with the chemical analysis (Table 6), showing that less yttrium is lost using a mechanically alloyed powder. Additionally, Figure 7a shows that similar nanoparticles can form in the SLM part consolidated with the unreinforced powder A. Other studies also observed such a population of nanoparticles containing alloying elements in small amounts (Such as Si or Mn) in unreinforced stainless steel produced by LAM [57]. It is likely that the population of nanoparticles observed in the different materials is constituted of different types of precipitate, oxygen getter elements (Y, Ti) forming in priority nano-oxides particles.

The observation of particles having an orientation relationship with the matrix (Figure 11) suggests that some of them can precipitate in a solid phase after solidification. The precipitation of ODS by LAM can therefore imply two complementary mechanisms. Initially, precipitation in the melt pool induces coarser particles due to the accelerated diffusion in a liquid phase. Their growth depends on the cooling rate. Secondly, a solid phase precipitation of the elements that has remained in solution in the matrix can occur due to the multiple thermal cycles in LAM that keep the solidified layers at high temperatures for a time.

In the literature, few authors mention precisely the densities of nano-oxides obtained in ODS—LAM, the best results are probably those recently obtained by Jia et al. with a density close to 4 × 10^21^ m^−3^ but oxides of 20 nm diameter on average [47]. For a volume fraction *f_v_* of added reinforcements equal between two materials, and assuming that it entirely corresponds to the spherical nano-precipitates population, a difference of one order of magnitude on the particle mean radius results in a difference of three orders of magnitude on the precipitates density. The mean radius of nano-oxides measured in conventional ODS steels is approximately 1 to 3 nm (as in the HIP material in this study), which corresponds to a difference of almost one order of magnitude from the mean radius measured on the LAM ODS materials.

Estimates of the nano-oxide density proposed in this study are consistent with this difference. Indeed, densities of nano-oxides of about 10^23^ to 10^24^ m^−3^ in conventional ODS steels are 2 to 4 orders of magnitude higher compared to the densities measured in the SLM ODS materials of this study (1.67 × 10^21^ m^−3^ with TEM characterizations, and 10^19^ to 10^20^ m^−3^ with SEM characterizations considering at best an interaction depth of backscattered electrons of ~100 nm). SAXS measurements confirm the low density of nano-oxides in SLM materials (<10^22^ m^−3^), both unreinforced and ODS (Figure 13).

Although the mass fraction of Y_2_O_3_ reinforcements was increased in powder SM-R0.5 (0.5% instead of 0.2% to 0.3% in conventional ODS steels), it enables the refinement of nano-particle size after SLM, the use of such powder does not enable a significant improvement in the nano-oxide density. However, the nanocomposite composite powder approach could be further explored using different types of reinforcing nanoparticles, different sizes of nanoparticles, or optimizing the powder elaboration process.

### 4.4. Influence of the Powder on the Tensile Properties

ODS steels obtained from powders SM-R0.5 and MA-R0.2 show very different mechanical properties (Figure 14). The brittle behavior obtained with ODS steels produced by SLM from a milled powder was also observed by Vasquez et al. [23]. These authors attribute this behavior to the coarse yttrium-rich phases that can act as crack initiation sites, also observed in this study (Figure 6). It is unclear why ODS steels consolidated from powder SM-R0.5 exhibit ductile behavior despite the presence of the same coarse phases and a similar microstructure. The lower laser scan speed used with powder MA-R0.2 may have led to larger Y-rich inclusions causing the brittle behavior. The large loss of yttrium after SLM with powder SM-R0.5 can also explain this ductile behavior close to the unreinforced steel consolidated with powder A.

Other authors also obtained a ductile tensile behavior with ODS ferritic steels consolidated by SLM or LMD from a milled powder [19,27]. They observe a slight increase in yield strength after a heat treatment at 1100 °C, which is attributed to fine precipitation of the yttrium remaining in a solid solution in the as-built material. Nevertheless, the ultimate strength of such LAM materials does not exceed at room temperature 700 MPa in literature for a ferritic ODS steel, which is still low compared to a simple HIP ODS (~900 MPa).

Ghayoor et al. observed a small increase of 35 MPa in yield strength with 304L ODS steel (obtained with a softly mixed powder) compared to 304L steel, both consolidated by SLM [44]. The results of our study do not show any increase in the mechanical tensile properties at room temperature between an ODS steel and an unreinforced steel produced by SLM. A slight increase in the elastic limit of 31 MPa and 25 MPa, respectively, at 650 °C and 700 °C is however observed, but remains far from the performance expected at these temperatures for an ODS steel.

In this study, ODS ferritic steel powders obtained by mechanical alloying or soft mixing do not lead to higher tensile properties than unreinforced ferritic steel when consolidated by laser additive manufacturing. The strengthening effect of the nanoparticle population is likely to be low and comparable to that of unreinforced steel. Adding Y_2_O_3_ satellite particles to a Fe-14Cr powder does not provide ODS steels with sufficient performance for high-temperature applications when consolidated by LAM. If some studies in the literature show slightly higher mechanical properties, they remain quite far from those obtained by conventional ODS metallurgy.

## 5. Conclusions

The aim of this work was to evaluate the fabricability of ferritic/martensitic ODS by laser additive manufacturing. For this purpose, two ODS powders were used; one from mechanical alloying and the other from a soft blend of steel powder with yttrium nano-oxides (nanocomposite powder). These two types of ODS powders are representative of the current powders used in LAM processes. It appears that:The use of a nanocomposite ODS powder enables the avoidance of mechanical alloying, and better flexibility in the choice of the reinforcement content or nature and in the choice of manufacturing parameters to achieve part density above 99%.Using a lower $\frac{P_{Laser}}{V_{Laser}}$ ratio, and therefore a higher cooling rate of the melt pool, enables the refinement of the nanoprecipitates.More than 80% of the yttrium is lost after SLM consolidation when using an ODS nanocomposite powder obtained by soft mixing. The nano-oxides are predominantly titanium oxides containing no or little yttrium. This is not the case with mechanically alloyed powder.The ODS steel consolidated by SLM from the nanocomposite powder has a mean radius of nanoparticles of approximately 8 nm, and their density is 1.67 × 10^21^ m^−3^ (TEM). SAXS measurements indicate that none of the SLM materials, ODS or unreinforced, contain a high density of fine nano-oxides comparable to those in conventional ODS.At room temperature, the yield strength of SLM parts consolidated from the nanocomposite ODS powder and the unreinforced powder is similar and around 400 MPa. No significant increase in tensile properties, 650 °C or 700 °C, is noticed between ODS steels and unreinforced ferritic steel consolidated by SLM.

The analysis of all the available data provided in this work and the literature shows that it is very difficult to obtain F/M ODS grades by LAM with the expected characteristics. Even if progress has been made using more adapted types of powder, the melting step strongly limits, for the moment, the possibility of obtaining fine and dense precipitation of nano-oxides. Other additive manufacturing technics avoiding fusion, such as cold spray or binder jetting, would be of particular interest for future work.

## Figures and Tables

**Figure 1 materials-16-02397-f001:**
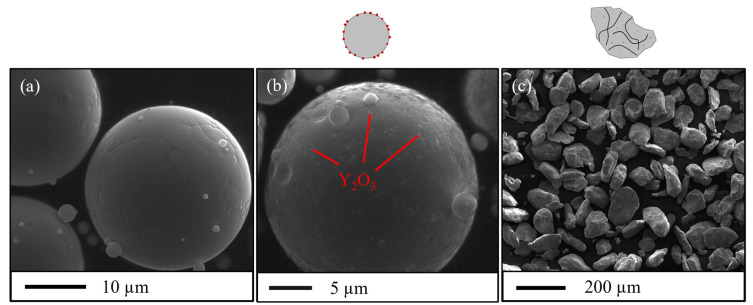
SEM images of the matrix Fe-14Cr-1W-0.22Ti powder A-Ti (**a**), of the «composite» ODS powder SM-R0.5 (**b**), and the mechanically alloyed ODS powder MA-R0.2 (**c**).

**Figure 2 materials-16-02397-f002:**
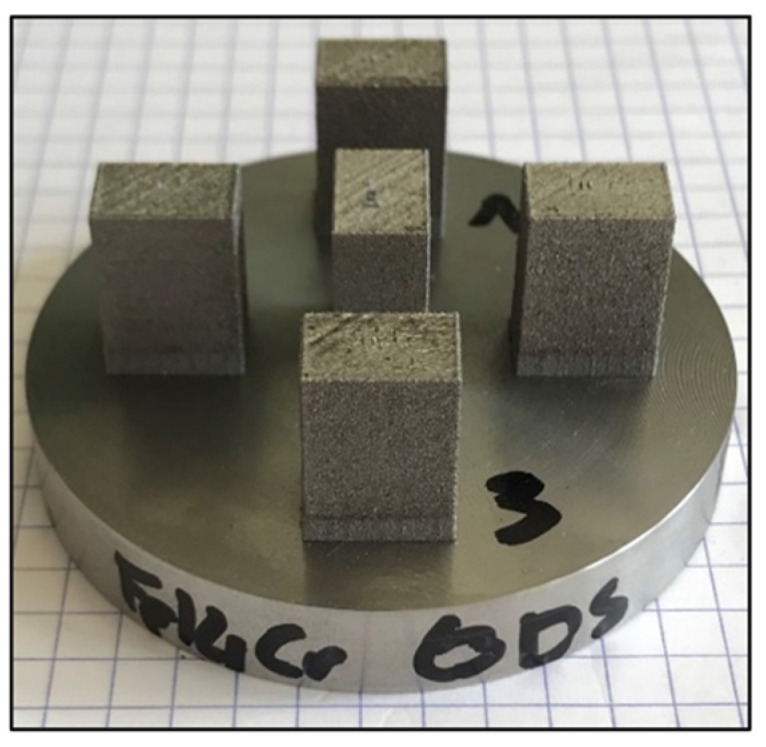
Picture of samples built on the 316L substrate plate using the ODS powder SM-R0.5.

**Figure 3 materials-16-02397-f003:**
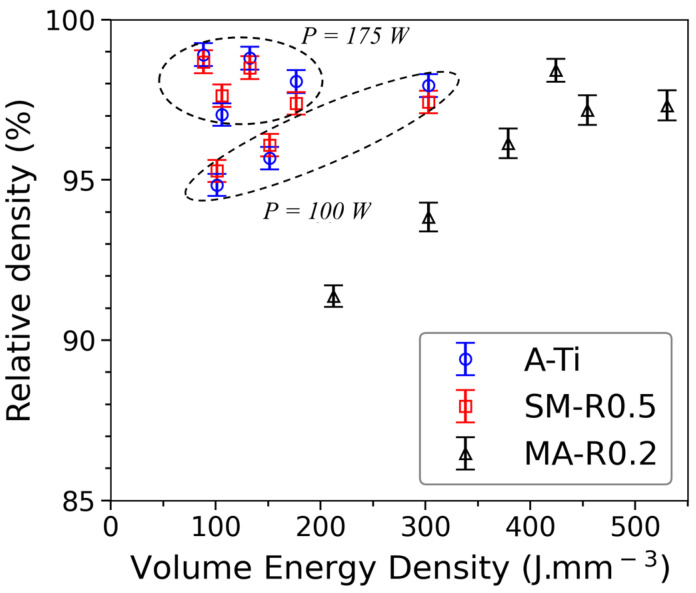
Relative density of the SLM consolidated samples, measured by Archimedes method, in relation to the volume energy density applied with powders A-Ti, SM-R0.5, and MA-R0.5.

**Figure 4 materials-16-02397-f004:**
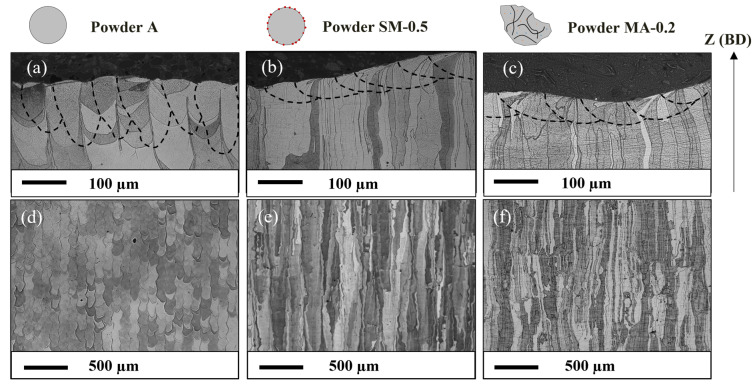
Optical images after etching showing the melt pool geometry and the microstructure along the build direction (BD) of SLM samples consolidated with powder A (**a**,**d**), ODS powder SM-R0.5 (**b**,**e**), ODS powder MA-R0.2 (**c**,**f**).

**Figure 5 materials-16-02397-f005:**
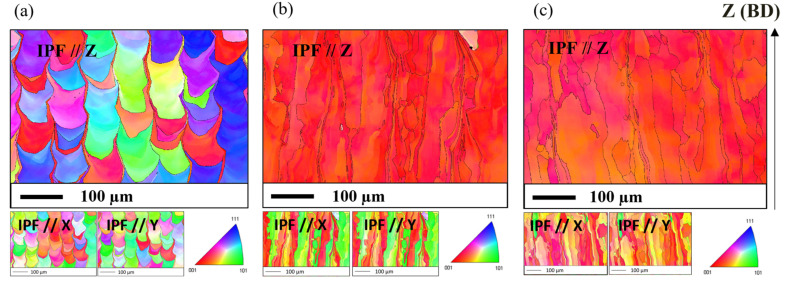
IPF maps obtained by EBSD of the microstructure along the build direction (BD) of SLM samples consolidated with powder A (**a**), ODS powder SM-R0.5 (**b**), and ODS powder MA-R0.2 (**c**).

**Figure 6 materials-16-02397-f006:**
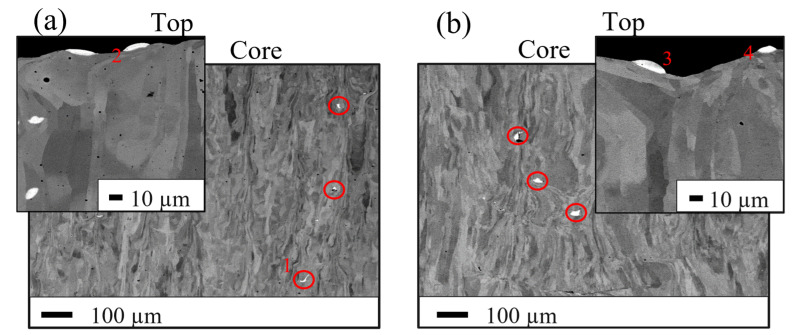
SEM-BSE images of SLM samples built with the ODS powder SM-R0.5 (**a**) and ODS powder MA-R0.2 (**b**). Red circles highlight the Y-rich coarse phases inside the parts, and red numbers identify particles on which EDX analysis is performed.

**Figure 7 materials-16-02397-f007:**
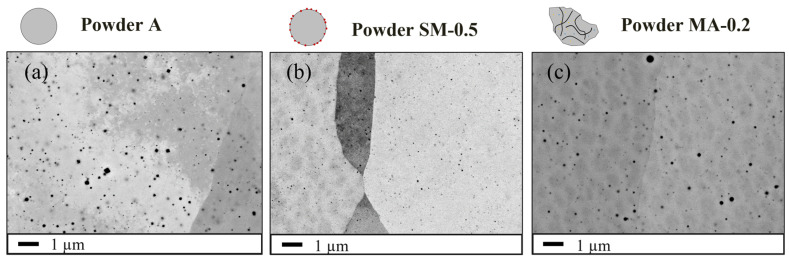
SEM-BSE images (×10 k) showing the nanoparticles distribution inside the microstructure of the SLM samples consolidated with powder A (**a**), ODS powder SM-R0.5 (**b**), and ODS powder MA-R0.2 (**c**).

**Figure 8 materials-16-02397-f008:**
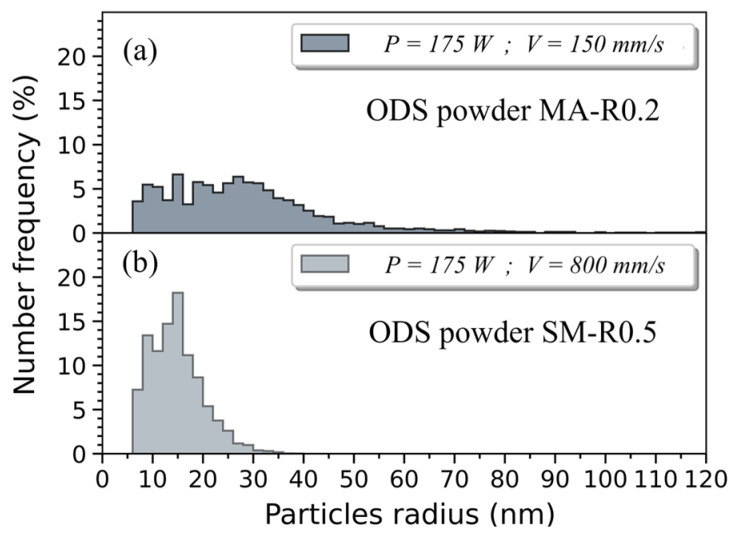
Particle size distribution measured with SEM images of SLM ODS samples built with powder SM-R0.5 (**a**) and powder MA-R0.2 (**b**).

**Figure 9 materials-16-02397-f009:**
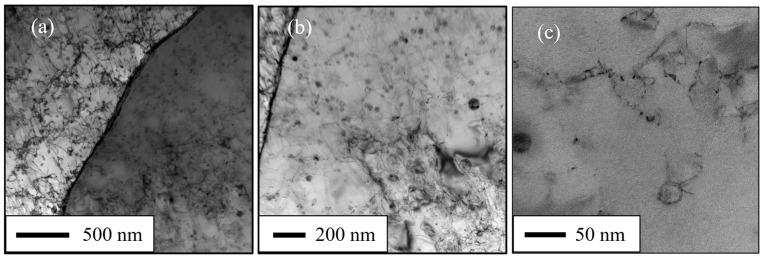
STEM images of the precipitation inside the SLM part consolidated with the ODS powder SM-R0.5 at magnifications (**a**) ×20 k, (**b**) ×30 k, and (**c**) ×150 k.

**Figure 10 materials-16-02397-f010:**
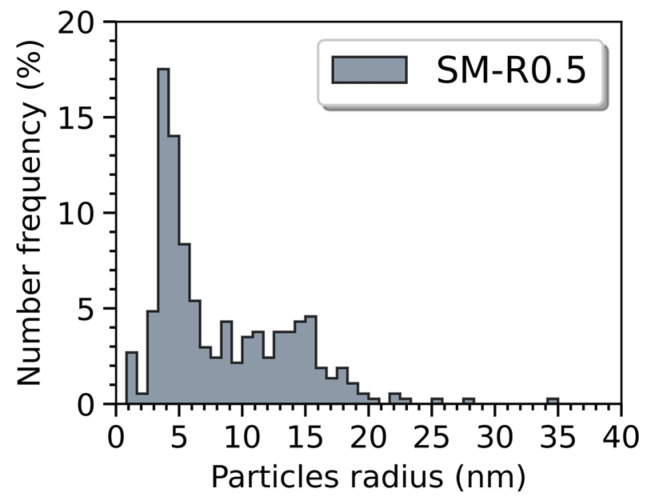
Particle size distribution measured with TEM images of SLM ODS samples built with powder SM-R0.5.

**Figure 11 materials-16-02397-f011:**
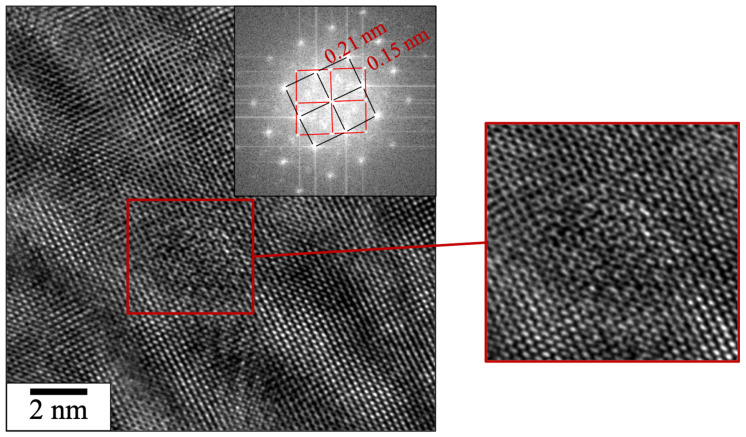
HR-TEM image of a 2 nm particle in the SLM sample built with composite ODS powder SM-R0.5.

**Figure 12 materials-16-02397-f012:**
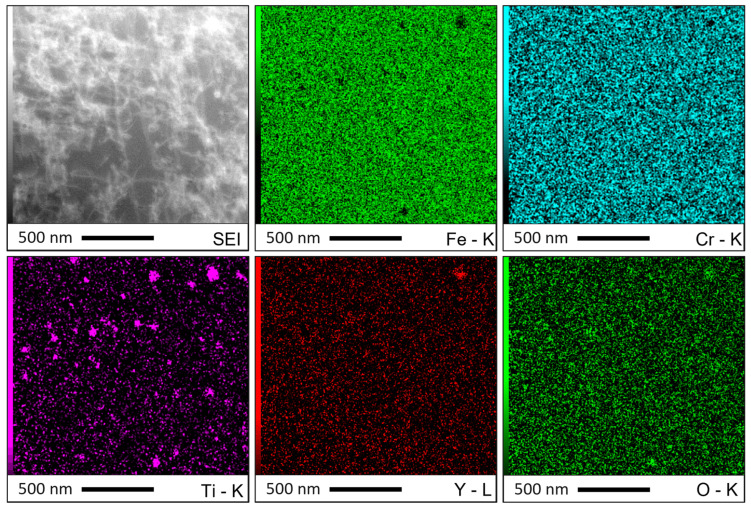
STEM EDX maps of the SLM sample built with composite ODS powder SM-R0.5. Each color map corresponds to an element and its characteristic spectral line.

**Figure 13 materials-16-02397-f013:**
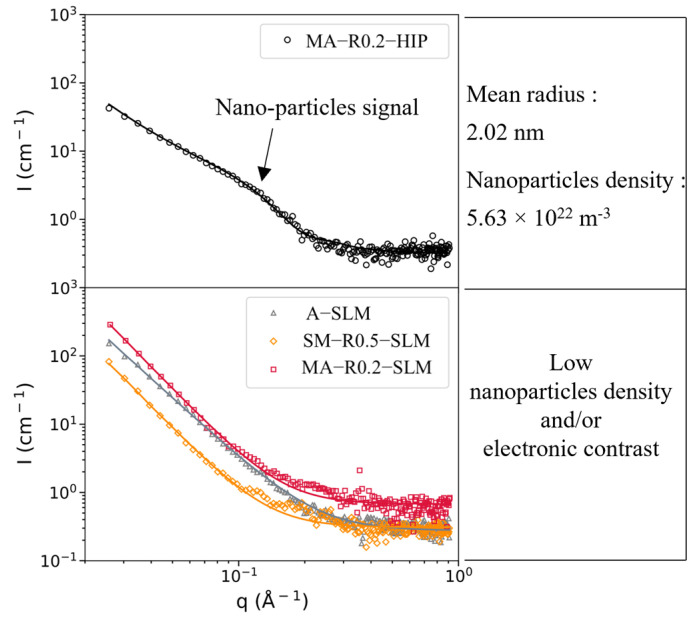
Experimental SAXS data and fits of the conventional HIPed ODS and the SLM materials.

**Figure 14 materials-16-02397-f014:**
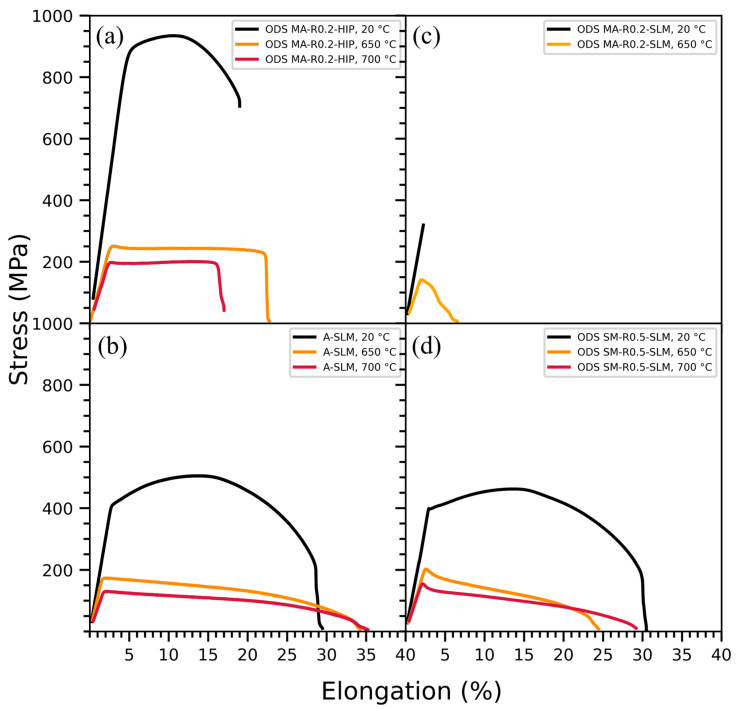
Tensile stress-strain curves at 20 °C, 650 °C and 700 °C of the (**a**) HIP ODS and the SLM samples built with powders (**b**) A, (**c**) MA-R0.2, and (**d**) SM-R0.5.

**Table 1 materials-16-02397-t001:** Chemical composition of powders. Values are given in %wt.

Element	Fe	Cr	Mn	Mo	Ni	W	C	Y	Ti	O	Y_2_O_3_ Target
A	Bal	14.05	0.07	0.145	0.292	0.96	0.038	-	-	0.093	0
A-Ti	Bal	14.2	0.06	<0.005	0.013	1.01	0.009	-	0.15	0.061	0
SM-R0.5	Bal	14.1	0.06	<0.005	0.012	1.01	0.015	0.29	0.15	0.21	0.5
MA-R0.2	Bal	13.5	0.092	<0.005	0.056	0.993	0.016	0.146	0.158	0.117	0.2

**Table 2 materials-16-02397-t002:** Powder size distribution characteristics.

Powder	D10 (µm)	D50 (µm)	D90 (µm)
A	8	22	60
A-Ti	10	20	43
MA	60	89	132

**Table 3 materials-16-02397-t003:** SLM window parameters applied to each powder.

Powder	Laser Power (W)	Scan Speed (mm.s^−1^)	Hatching Distance (µm)	Layer Thickness (µm)	VED (J.mm^−3^)
14Cr atomized (A, A-Ti, SM-R0.5)	100–175	200–1200	65–120	30	88–303
14Cr MA (MA-R0.2)	100–175	150–300	90	20–50	212–530

**Table 4 materials-16-02397-t004:** Melt pool dimensions in samples consolidated with each powder.

Sample	A-SLM	SM-R0.5-SLM	MA-R0.2-SLM
Melt pool width (µm)	110 ± 6.5	205 ± 14.7	206 ± 10.2
Melt pool depth (µm)	140 ± 5.7	50 ± 2.6	69 ± 4.1

**Table 5 materials-16-02397-t005:** EDX chemical analysis results on coarse particles indicated in Figure 6. Values are given in %wt.

EDX Spectrum on Particle n°	Y	Ti	O	Fe	Cr	C	Al
1	89.05	3.37	4.01	2.45	1.11	0	0
2	93.14	1.04	3.8	1.16	0	0.86	0
3	87.78	5.65	4.49	1.3	0.36	0	1.01
4	87.09	5.08	4.15	1.81	1.31	0	0.55

**Table 6 materials-16-02397-t006:** Chemical analysis on SLM parts consolidated from powders A -SLM, SM-R0.5-SLM, and MA-R0.2-SLM. Initial yttrium and oxygen content in powders are reminded.

	SLM part			Powder	
Elements	Ti	Y	O	Y	O
A -SLM	<0.15	NA	0.16	NA	0.093
SM-R0.5-SLM	0.14	0.05	0.068	0.29	0.21
MA-R0.2-SLM	0.16	0.12	0.17	0.146	0.117

## Data Availability

Data will be made available on request.

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
