# Peer review of "Assessment of Ferritic ODS Steels Obtained by Laser Additive Manufacturing"

_materials, 2023, doi:10.3390/ma16062397_

Round 1

Reviewer 1 Report

The manuscript entitled “materials-2250111-Laser AM” dealing with laser additive manufacturing has been reviewed. The paper has been nicely written but needs significant improvement. Please follow my comments.

1.     What is the main research question for this research work?

2.     Please check the chemical composition in Table 1.

3.     What is the future direction of this work?

4.     How did you select the parameters?

5.     Please update the introduction with the new publications in the field. Authors are encouraged to read and add the following two new papers in the field.

·       Microstructure simulation and experimental evaluation of the anisotropy of 316 L stainless steel manufactured by laser powder bed fusion

·       Process-dependent anisotropic thermal conductivity of laser powder bed fusion AlSi10Mg: impact of microstructure and aluminum-silicon interfaces

6.     Please proofread the paper.

7.     Add more discussion about “Figure 12. STEM EDX maps of the SLM sample”.

8.     Additive manufacturing has many advantages over the conventional manufacturing method which can be highlighted in your paper. Please read the following manuscript and add it to the literature to show how additive manufacturing is comparable with conventional manufacturing.

·       Laser subtractive and laser powder bed fusion of metals: review of process and production features

Reviewer 2 Report

well done

Reviewer 3 Report

The paper is well presented and contains original results. However, authors are encouraged to improve their work based on the following comments:

1) The novelty of this work must be more explained.

2) The applications of this work should be more discussed.

3) For general readers, authors are encouraged to discuss other kind of works such as: [(a) “Microstructural/geometric imperfection sensitivity on the vibration response of geometrically discontinuous bi-directional functionally graded plates (2D-FGPs) with partial supports by using FEM”, Steel and Composite Structures, 45(5), 621-640.; (b) “Combined influence of porosity and elastic foundation parameters on the bending behavior of advanced sandwich structures”, Steel and Composite Structures, 46(1), 1-13.].

4) In conclusion, give only main findings of your research with an appropriate value.
